# SEGHI Study: Defining the Best Surveillance Strategy in Hodgkin Lymphoma after First-Line Treatment

**DOI:** 10.3390/cancers13102412

**Published:** 2021-05-17

**Authors:** Mariana Bastos Oreiro, Reyes Martín, Pilar Gomez, Nieves López Muñoz, Antonia Rodriguez, Marta Liébana, Belén Navarro, Blanca Sánchez-González, Pilar Marí, Jaime Pérez de Oteiza, Antonio Gutiérrez, Leyre Bento, Eva Domingo Doménech, María Jesús Vidal, Raquel Del Campo, Elena Pérez Ceballos, María Infante, Alicia Roldán, Daniel García Belmonte, Miriam Santero, Anna Sureda, Ramón García Sanz

**Affiliations:** 1Department of Hematology, Gregorio Marañon Health Research Institute, Gregorio Marañón Hospital, 28007 Madrid, Spain; reyes.martinrojas@gmail.com; 2Hematology Department, Hospital La Paz, 28046 Madrid, Spain; pilar.gph@gmail.com; 3Hematology Department, Hospital 12 de Octubre, 28041 Madrid, Spain; nieves_92@yahoo.com (N.L.M.); antonia.rodriguez@salud.madrid.org (A.R.); 4Hematology Department, Hospital Puerta de Hierro, 28222 Madrid, Spain; martalvillela@gmail.com (M.L.); mariabelen.navarro@salud.madrid.org (B.N.); 5Hematology Department, Hospital del Mar, 08003 Barcelona, Spain; bsanchezgonzalez@parcdesalutmar.cat; 6Hematology Department, Hospital HM Madrid Sanchinarro, Universidad CEU San Pablo, 28050 Madrid, Spain; mpmari@hmhospitales.com (P.M.); jperezoteyza@hmhospitales.com (J.P.d.O.); 7Hematology Department, Hospital Universitario Son Es pases (IdiSBa), 07120 Palma de Mallorca, Spain; antoniom.gutierrez@ssib.es (A.G.); leyrebento@gmail.com (L.B.); 8Hematology Department, Hospital ICO, 08908 Barcelona, Spain; edomingo@iconcologia.net (E.D.D.); asureda@iconcologia.net (A.S.); 9Hematology Department, Hospital Universitario de León, 24071 León, Spain; mjvidalm2010@gmail.com; 10Hematology Department, Hospital Universitario Son Llátzer, 07198 Palma de Mallorca, Spain; rcampo@hsll.es; 11Hematology Department, Hospital General Universitario Morales Meseguer, 30008 Murcia, Spain; epceballos@gmail.com; 12Hematology Department, Hospital Infanta Leonor, 28031 Madrid, Spain; mariastefania.infante@salud.madrid.org; 13Hematology Department, Hospital Infanfa Sofía, 28702 Madrid, Spain; aroldanp@salud.madrid.org; 14Hematology Department, Hospital de La Zarzuela, 28023 Madrid, Spain; dgarciabe@sanitas.es; 15Hematology Department, Hospital de Torrejón, 28850 Torrejón de Ardóz, Spain; msantero@torrejonsalud.com; 16Hematology Department, Hospital Clínico de Salamanca, 37008 Salamanca, Spain; rgarcias@usal.es

**Keywords:** Hodgkin lymphoma, surveillance, follow-up, images

## Abstract

**Simple Summary:**

As the optimal strategy for early surveillance after first complete response is unclear in Hodgkin lymphoma (HL), with the aim to identify the best one, we analyzed surveillance strategies after first-line treatment in a group of 640 Hodgkin lymphoma patients from 15 different GELTAMO centers in Spain. In this study, we have demonstrated that surveillance approaches with imaging in HL do not provide benefits for patient survival, nor do they anticipate relapse. Our results are of great importance, because they could contribute to improve the follow up after first complete response of this group of patients.

**Abstract:**

The optimal strategy for early surveillance after first complete response is unclear in Hodgkin lymphoma. Thus, we compared the various follow-up strategies in a multicenter study. All the included patients had a negative positron emission tomography/computed tomography at the end of induction therapy. From January 2007 to January 2018, we recruited 640 patients from 15 centers in Spain. Comparing the groups in which serial imaging were performed, the clinical/analytical follow-up group was exposed to significantly fewer imaging tests and less radiation. With a median follow-up of 127 months, progression-free survival at 60 months of the entire series was 88% and the overall survival was 97%. No significant differences in survival or progression-free survival were found among the various surveillance strategies. This study suggests that follow-up approaches with imaging in Hodgkin lymphoma provide no benefits for patient survival, and we believe that clinical/analytical surveillance for this group of patients could be the best course of action.

## 1. Introduction

Hodgkin lymphoma (HL) has a high cure rate, close to 80% [1,2,3]. Thus, a large percentage of patients who are followed up will not relapse. The optimal strategy is unclear for early follow-up in HL when complete response (CR) is achieved after first-line treatment. In general, however, surveillance strategies are increasingly oriented toward a reduced number of radiological studies.

Few retrospective studies have analyzed the optimal strategy in this setting; however, most were performed before CT imaging was included in routine practice [4,5]. Notably, these studies suggest that symptoms are the most important factor for monitoring these patients, based on the fact that relapses are usually symptomatic and mainly occur during the first two years of follow-up.

Imaging follow-up is not generally considered cost-effective to detect relapses [6,7] of lymphomas. On the other hand, some studies have suggested that anticipating relapse does not translate into a real benefit in terms of patient survival [8]. Along these lines, some cost-effectiveness studies indicated that strategies including radiological images in the surveillance of this pathology are not cost-effective and could be detrimental to the patient’s quality of life [8,9,10]. El Galay et al. reported that positron emission tomography/computed tomography (PET/CT) surveillance, both routine and clinically indicated, is associated with a low positive predictive value (PPV) and that routine PET/CT surveillance has unacceptably high costs [9]. Imaging-based surveillance has also led to concerns about the potential long-term biologic effects of ionizing radiation, particularly in young patients with HL who have a high probability of long-term survival [11].

Nevertheless, this issue is not entirely clear, and recommendations in this regard contradict each other. Entities such as the American Society of Hematology Choosing Wisely and the European Society for Medical Oncology suggest that it is unnecessary to perform follow-up with imaging in these patients [12]. However, there are international guidelines based on expert recommendations, such as those of the National Comprehensive Cancer Network, which find it appropriate to perform a control CT scan at 6, 12, and 24 months after the end of treatment [13].

For this reason, the Spanish lymphoma/autologous marrow transplant study group (Grupo Español de Linfomas/Trasplante Autólogo de Médula Ósea [GELTAMO]) performed the SEGHI study (“Seguimiento en Hodgkin por imágenes”), with the intention to retrospectively analyze the various follow-up strategies in multiple GELTAMO centers after first CR in patients diagnosed with HL.

## 2. Materials and Methods

This was a retrospective, multicenter study performed by the GELTAMO group between January 2007 and January 2018, which included consecutive patients with a diagnosis of classical HL who had a negative PET/CT at the end of induction (Deauville score 1–3) [14]. With an assumed equal efficacy of both arms (clinical and image follow up), a hypothetical inferiority of clinical surveillance with an overall survival rate of 80% versus a rate of 90% or more with image (corresponding to a non-inferiority margin of 10%) had to be excluded with 95% confidence and a power of 80%. Therefore, 200 patients were needed per group. Patients older than 18 years with refractory disease, who had not reached CR at the end of first-line treatment, or in whom the complete response had been assessed only by CT scan were excluded. Demographic characteristics, disease characteristics, and biological and imaging data were collected through a centralized digital platform. As is typical, the disease stage was categorized as localized (I–II) or advanced (III–IV) [14,15]. For the analysis, each center defined the strategy used, at least for the first two years of follow-up. The patients were categorized into five groups according to the strategy used by each doctor at the referral center: Clinical group (C/A) (follow-up by medical history, laboratory tests, and physical examination); CT-3 (CT every three months); CT-6 (CT every six months); PET CT-3 (PET/CT every three months); and PET CT-6 (PET/CT every six months). For each individual patient, each researcher identified the strategy used. The dose used to calculate the radiation received for each group was 19.98 mSv for each body CT and 6.7 mSv for each PET/CT (European Commission. Radiation Protection No 180. Medical radiation exposure of the European population. EU, 2014). The time to relapse and mortality were compared between the groups, as were the false suspicion rates, the number of images taken per patient, and the accumulated radiation.

## 3. Results

Until June 2020, 707 patients were recruited from 15 GELTAMO centers across Spain, of which 57 were excluded for not having a CR according to PET/CT at the end of induction or progression within three months since the final PET/CT induction. Ultimately, 640 patients were analyzed. The patients’ characteristics as well as the data related to the disease and the treatment received are shown in Table 1. A CT scan every six months was the most common surveillance strategy, comprising 232 patients (36.3%), followed by the clinical/analytical strategy, with 202 patients (31%). Table 2 summarizes the distribution by lymphoma and treatment characteristics among the various surveillance strategies. As can be observed, in the groups followed up by CT-3 and PET CT-6 imaging, a greater number of patients with advanced stages and high international prognostic index are grouped. Table 3 shows the number of visits and imaging performed employing each surveillance strategy, as well as the median radiation dose received in each group. As expected, compared with the groups in which serial imaging was performed, the C/A follow-up group was significantly exposed to fewer imaging tests and less radiation.

Sixty-eight patients relapsed, 22 of whom had a localized stage. In 61.8% of patients, regardless of the follow-up group to which patients belonged, the suspicion of relapse was clinical (Table 4). Medical history and physical examination were the most common ways of identifying relapse. Laboratory abnormalities infrequently identify relapses (in just 2.9% of patients as an isolated finding and in 18% when combined with clinical data or image findings). An abnormal blood count was only identified in relapses at advanced stages (six cases) and in none of the localized ones. High lactate dehydrogenase (LDH) (four cases) and erythrocyte sedimentation rate (ESR) (six cases) were also more frequent in the advanced stages (75% and 66%, respectively). Of the 44 relapses that occurred in the imaging follow-up groups, only 26 were identified by imaging, with 18 (40%) identified by clinical or laboratory data. In 62% of the cases, relapse was in an accessible location by physical examination, such as those in the cervical, supraclavicular, axillary, and inguinal regions.

The PPV was 59% for CT, 47% for PET/CT, and 64% for C/A follow-up.

With a median follow-up of 127 months, progression-free survival (PFS) of the entire series at 60 months was 88% (95% confidence interval (CI) 0.84–0.90) and the overall survival (OS) 97% ((95%CI 0.95–0.98), 99% (I95%CI 0.95–0.99) for localized stages and 94% (95%CI 0.90–0.96) for advanced stages) (Figure 1). Figure 2 shows the OS and Figure 3 shows the PFS for each follow-up group. The OS was 96% (95%CI 0.90–0.98) for C/A, 95% (95%CI 0.90–0.98) for CT-3, 97% (95%CI 0.93–0.99) for CT-6, 96% (IC95 0.80–0.99) for PET/CT-3, and 93% (IC95 0.89–0.97) for PET/CT-6. Grouping the imaging strategies (Figure 2B), the OS for CT was 97% (95%CI 0.95–0.98) and for PET/CT, 96% (95%CI 0.90–0.98). In Figure 4, the same analysis was performed according to localized or advanced stage and no significant differences in survival were found among the surveillance strategies. The time to relapse and PFS for the various groups at 12, 24, and 60 months is described in Table 5.

## 4. Discussion

In this study, the usefulness of imaging follow-up for patients with HL who had achieved CR at the end of first-line induction treatment was analyzed in a large number of patients with a long-term follow-up of more than 10 years. In this regard, we have found that the clinical/analytical surveillance strategy achieves results very similar to the imaging follow-up, avoiding systematic radiation and unnecessary direct and indirect costs. We have observed that in most cases, relapse can be identified through the patient’s medical history or physical examination. Of note, even in the groups that were followed-up with imaging, in 40% of patients, relapse was identified by clinical or analytical abnormalities. In addition, in more than half of the cases included in the study, the lesion identified in the image was accessible to physical examination, although it is possible that those clinicians who performed an imaging follow-up probably performed a less extensive physical examination. Abnormalities in the blood tests were infrequent, six as an identifier of relapse in our study, with only 18% of patients in whom a relapse was suspected due to the presence of these findings. The erythrocyte sedimentation rate has been previously identified as a marker of early relapse [16,17], but it infrequently detected relapse in our study. The fact that we excluded primarily refractory patients and that less attention might have been paid to the analytical data of the groups that underwent imaging could explain this difference. Of the relapsed patients, however, only seven presented these abnormal data.

Although some guidelines recommend CT at 6 and 24 months in these patients, there is no robust evidence supporting this strategy [13]. In fact, there is evidence that supports clinical follow-up over imaging follow-up. In a study including 78 patients with localized-stage classical HL treated with doxorubicin, bleomycin, vinblastine, and dacarbazine, the relapse rate for this group was very low and they found 11 cases of false positives, concluding that elimination of surveillance imaging could reduce healthcare expenses and cumulative radiation doses [18]. Another recent study, which included 179 patients with classical HL at all stages who reached CR at the end of first-line treatment, identified that only 5% of the patients relapsed, for whom 463 scans were performed per relapse detected in the two-year post-therapy surveillance [19]. Likewise, hypothetical cost-benefit analyses have demonstrated CT follow-up to be of minimal survival benefit for patients at all stages of disease and to be associated with a reduction in quality-adjusted life expectancy for those with early-stage disease [8].

PET CT is the best choice to detect disease in HL [14,20], given that compared with CT, PET CT is better for distinguishing between residual lesions, necrosis, and active disease [20,21]. There is even clear evidence that, at the end of induction chemotherapy in patients with advanced disease, PET CT allows us to identify those patients in whom the disease remains active and who will have a high probability of relapse [22,23]. However, it is important to note that nearly 40% of patients with a positive PET CT will not relapse [9,24]; therefore, using PET CT as a follow-up strategy will generate more subsequent studies, more imaging, and unnecessary stress for the patients. In agreement with these studies, follow-up was 47% in our analysis the PPV for the PET CT, which means that more than half of the suspicions of relapse in this group were not ultimately confirmed as such. In addition, we have shown in this study that there are no significant differences in the time to relapse or, more importantly, in overall survival, regardless of the follow-up strategy used. It is important to clarify that the drop observed in the PET 7 CT follow-up curve has to do with three deaths that occurred due to causes unrelated to the disease in no-relapse patients. On the other hand, all the strategies include cases of suspected relapse that are later not confirmed. Thus, even in the C/A strategy, the images are generated that later will not show relapse. However, the C/A follow-up strategy has shown the highest PPV in our study, with a notably smaller number of images and less exposure to radiation. It is striking that the advanced CT-3 group appeared to have a greater number of relapses. This result could have to do with a bias in the study, by which it was decided to perform this surveillance strategy in patients with more aggressive characteristics. Although we consider this option as the most likely, it should be noted that with the study design, it cannot be ruled out that the CT3 strategy manages to anticipate the identification of relapse, and this could only be elucidated with a comparative prospective study. Moreover, as we demonstrated in this analysis, follow up with CT-3 contributed the greatest cumulative dose of radiation, with a median cumulative dose per patient of more than 140 msV in five years, and for patients followed up with CT-6, 100 msV in five years. Considering that the excess relative risk for cancers other than leukemia was estimated to be 0.97 (0.14 to 1.97) per Sv and the excess relative risk for leukemia excluding chronic lymphocytic leukemia was estimated to be 1.93 per Sv (<0 to 8.37) [25], the risk to which we are exposing our patients is not negligible. On the opposite side, a benefit that imaging follow-up could provide would be the detection of secondary neoplasms or late toxicities related to treatment. This aspect was not analyzed in our studies. Finally, it is also important to mention that ultrasonography, whose utility has not been evaluated in this study, can be a useful tool in the identification of several lymph nodes without providing radiation.

## 5. Conclusions

In conclusion, to our knowledge, this is the largest study to date that compares surveillance strategies in patients with classical HL. We have demonstrated that follow-up approaches with imaging in HL do not provide benefits for patient survival, nor do they anticipate relapse, which generates unnecessary risk for these patients in addition to excessive costs for the healthcare system. Follow-up with a medical history and a physical and analytical examination is probably the best follow-up strategy for these patients and we recommend this practice.

## Figures and Tables

**Figure 1 cancers-13-02412-f001:**
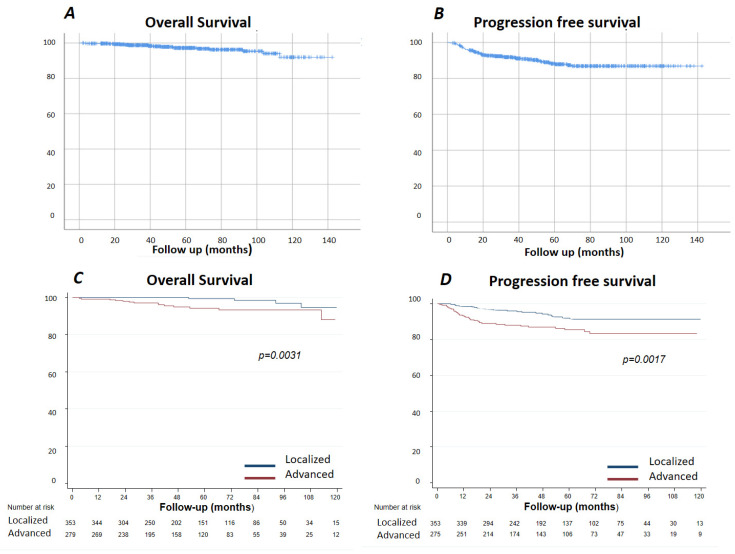
Overall survival and progression-free survival for the whole series and by stage. (**A**) Overall Survival. (**B**) Progression free survival. (**C**) Overall Survival by stage. (**D**) Progression free survival by stage.

**Figure 2 cancers-13-02412-f002:**
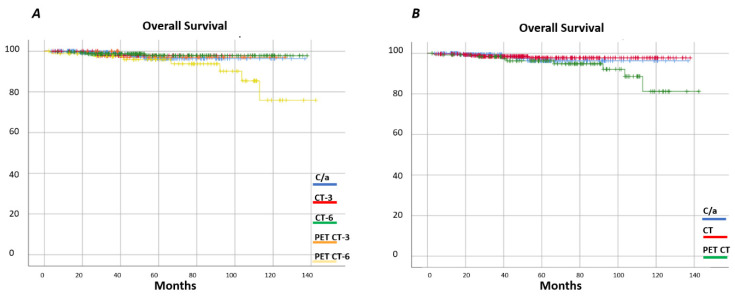
Overall survival by surveillance group. (**A**) OS by clinical analytical, three-month CT, six-month CT, three-month PET CT, and six-month PET CT surveillance strategy; (**B**) OS by clinical analytical, CT, and PET CT surveillance strategy.

**Figure 3 cancers-13-02412-f003:**
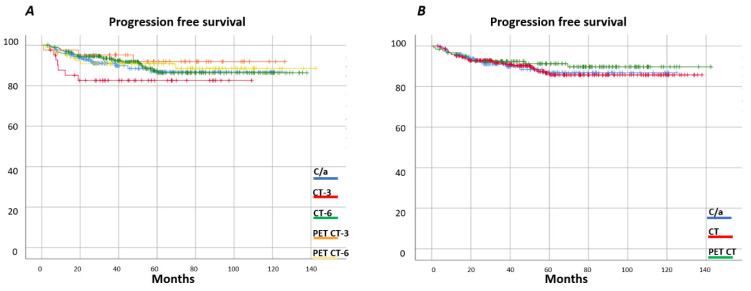
Progression-free survival by surveillance group. (**A**) OS by clinical analytical, three-month CT, six-month CT, three-month PET CT, and six-month PET CT surveillance strategy; (**B**) OS by clinical analytical, CT, and PET CT surveillance strategy.

**Figure 4 cancers-13-02412-f004:**
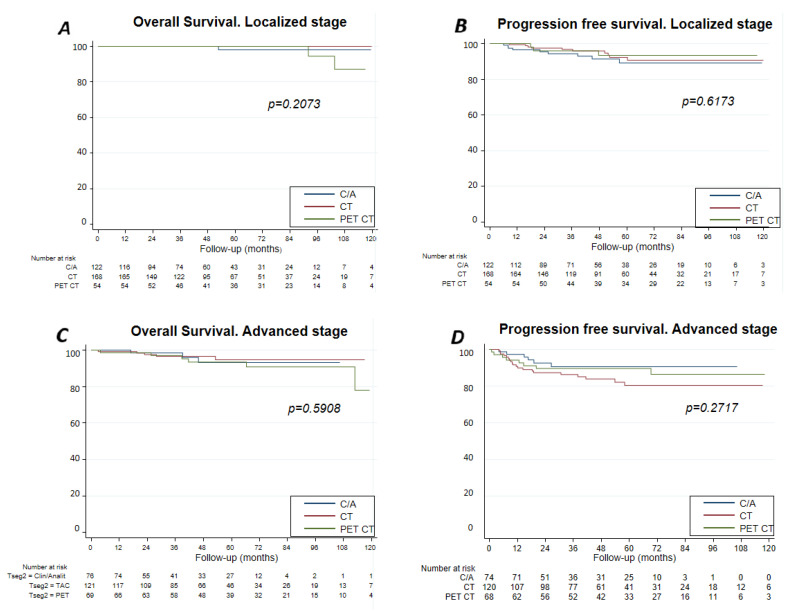
Overall survival and Progression-free survival by surveillance group considering the stage. (**A**,**B**) Overall survival and progression-free survival by surveillance group by stage. (**C**,**D**) OS by surveillance for localized stages; by surveillance for advanced stages.

**Table 1 cancers-13-02412-t001:** General characteristics of the patients.

Characteristics	*N* = 640 Patients
Age (median)	45.5 years (r: 18.2–93.2)
Histological Type	NSHL: 434 (67.8%) MCHL: 112 (17.5%) LRHL: 44 (6.9%) LDHL: 5 (0.8%) Not specified: 45 (7%)
Stage	I–II: 375 (58.6%) III–IV: 262 (41%) Missing data: 3 (0.4%)
Treatment Group	Favorable localized: 178 (27.8%) Unfavorable localized: 179 (28%) Advanced IPS 1–3: 166 (25.9%) Advanced IPS 4–7: 115 (18%) Missing data: 2 (0.3%)
Type of treatment	Radiotherapy: 35 (5.5%) Radiotherapy + chemotherapy: 253 (39.5%) Chemotherapy: 350 (54.7%) Missing data: 2 (0.3%)
Chemotherapy	ABVD: 564 (88.1%) BEACOPP: 37 (5.8%) Other: 29 (4.5%) Missing data: 10 (1.6%)
Follow-up strategies	Clinical/analytical: 202 (31.6%) CT-Scan every 3 months: 58 (9.1%) CT- Scan every 6 months: 232 (36.3%) PET/CT every 3 months: 43 (6.7%) PET/CT every 6 months: 82 (12.8%) Missing data: 23 (3.5%)

NSHL: Nodular sclerosis Hodgkin lymphoma; MCHL: Mixed cellularity Hodgkin lymphoma; LRHL: Lymphocyte-rich Hodgkin lymphoma; LDHL: Lymphocyte- depleted Hodgkin lymphoma; IPI: International Prognostic Index; ABVD: Adriamycin, Bleomycin, Vinblastine, Dacarbazine; BEACOPP: Bleomycin, Etoposide, Adriamycin, Cyclophosphamide, Vincristine, Procarbazine, Prednisone.

**Table 2 cancers-13-02412-t002:** General characteristics of the patients presented by follow-up strategy.

		Follow-Up Strategy				
		Clinical/Analytical (*n* = 202)	CT-Scan Every 3 Months (*n* = 58)	CT-Scan Every 6 Months (*n* = 232)	PET/CT Every 3 Months (*n* = 43)	PET/CT Every 6 Months (*n* = 82)
**Age (median)**		46.9 (r:18.2–93.2)	41.2 (r:20.7–84.7)	45.6 (r:20.4–92.4%)	44.2 (r:19.8–82.7)	45.7 (r:20.1–90.2)
Histological Type	NSHL	125 (63.5%)	47 (81%)	149 (64.8%)	39 (90.7%)	58 (75.3%)
	MCHL	34 (17.3%)	8 (13.8%)	49 (21.3%)	2 (4.7%)	16 (20.8%)
	LRHL	20(10.2%)	1 (1.7%)	17 (7.4%)	1 (2.3%)	3 (3.9%)
	LDHL	1 (0.5%)	0 (0%)	3 (1.3%)	0 (0%)	0 (0%)
	Not specified	17 (8.5%)	2 (3.4%)	12 (5.2%)	1 (2.3%)	0 (0%)
Stage	I-II	130 (64.7%)	32 (55.1%)	141 (60.8%)	19 (44.2%)	44 (54.3%)
	III-IV	71 (35.3%)	26 (44.8%)	91 (39.2%)	24 (55.8%)	37 (45.7%)
Treatment Group	Favorable localized	68 (33.8%)	23 (39.7%)	61 (26.3%)	3 (7%)	20 (24.4%)
	Unfavorable localized	57 (28.4%)	8 (13.8%)	76 (32.8%)	13 (30.2%)	19 (23.2%)
	Advanced IPI 1–3	46 (22.9%)	10 (17.2%)	68 (29.3%)	17 (39.5%)	17 (20.7%)
	Advanced IPI 4–7	30 (14.9%)	17 (29.3%)	27 (11.6%)	10 (23.3%)	26 (31.7%)
Type of treatment	Radiotherapy	6 (3%)	13 (22.4%)	15 (6.5%)	1 (2.4%)	0 (0%)
	Chemo+Radiotherapy	74 (36.6%)	28 (48.3%)	112 (48.3%)	7 (16.7%)	24 (29.3%)
	Chemotherapy	122 (60.4%)	17 (29.3%)	105 (45.3%)	34 (81%)	58 (70.7%)
Type of chemotherapy	ABVD	171 (86.8%)	56 (98.2%)	217 (94.3%)	39 (92.9%)	59 (72%)
	BEACOPP	12 (6.1%)	0 (0%)	3 (1.3%)	3 (7.1%)	19 (23.2%)
	Other	14 (7.1%)	1 (1.8%)	10 (4.3%)	0 (0%)	4 (4.9%)

NSHL: Nodular sclerosis Hodgkin lymphoma; MCHL: Mixed cellularity Hodgkin lymphoma; LRHL: Lymphocyte-rich Hodgkin lymphoma; LDHL: Lymphocyte- depleted Hodgkin lymphoma; NLPHL: Nodular lymphocyte-predominant Hodgkin lymphoma; IPI: International Prognostic Index; ABVD: Adriamycin, Bleomycin, Vinblastine, Dacarbazine; BEACOPP: Bleomycin, Etoposide, Adriamycin, Cyclophosphamide, Vincristine, Procarbazine, Prednisone.

**Table 3 cancers-13-02412-t003:** Number of imaging tests based on follow-up strategy.

	Follow-Up Strategy					*p*-Value
	Clinical/Analytical (*n* = 202)	CT-Scan Every 3 Months (*n* = 58)	CT-Scan Every 6 Months (*n* = 232)	PET/CT Every 3 Months (*n* = 43)	PET/CT Every 6 Months (*n* = 82)	
Visits/year (median)	4 (0–10)	4 (2–12)	4 (1–15)	6 (2–21)	4 (1–12)	<0.001 *
Median number of CT-scans (2y) (*n* = 579)	1 (0–6)	6 (2–10)	3 (2–8)	2 (0–4)	0 (0–4)	0.001 *
Median number of CT-scans (5y) (*n* = 319)	3 (0–17)	7 (3–14)	5 (2–11)	3 (0–7)	0 (0–5)	<0.001 *
Median number of PET/CT (2y) (*n* = 579)	0 (0–8)	1 (1–6)	0 (0–6)	6 (1–10)	3 (1–9)	<0.001 *
Median number of PET/CT (5y) (*n* = 319)	0 (0–5)	1 (1–7)	0 (0–6)	8 (3–16)	6 (2–9)	<0.001 *
2-year Cumulative Radiation Exposure (msV) (median) (*n* = 579)	19.9 (0–140)	126.68 (46.68–240.08)	59.94 (0–200.04)	80.16 (6.7–187.12)	20.1 (13.4–140.22)	<0.001 *
5-year Cumulative Radiation Exposure (msV) (median) (*n* = 319)	40.1 (0–339.7)	146.56 (66.64–326.62)	99.9 (39.9–259.98)	113.54 (20.1–206.86)	40.2 (13.4–160.2)	<0.001 *

msV: millisievert. * *p* ≤ 0.001.

**Table 4 cancers-13-02412-t004:** Characteristics of relapse detection.

Findings that Identified Relapse	Relapse (*n* = 68)
Clinical History	9 (13.2%)
Physical Examination	10 (14.7%)
Laboratory abnormalities	2 (2.9%)
Clinical History + Physical Examination	5 (7.4%)
Clinical History + Physical Examination + Laboratory abnormalities	8 (11.8%)
Clinical/ analytical + image	8 (11.8%)
	42 (61.7%)
CT-SCAN	17 (25.1%)
PET/CT	9 (13.1%)
	26 (38.2%)
Type of laboratory finding	
Blood cell counts	6 (33.3%)
ESR	7 (38.8%)
LDH	4 (22.2%)
Other	1 (5.6%)

CT-SCAN: computer tomography scan. PET/CT: positron emission tomography/computed tomography LDH: Lactate Dehydrogenase. ESR: Erythrocyte sedimentation rate.

**Table 5 cancers-13-02412-t005:** Progression free survival and time to relapse.

Surveillance Strategies	PFS at 12 Months	PFS at 24 Months	PSF at 60 Months	Median Time to Relapse (Month)
Clinical/analytical (*n* = 202)	96%	94%	86%	16.2
CT-Scan every 3 months (*n* = 58)	85%	82%	82%	9.4
CT-Scan every 6 months (*n* = 232)	96%	96%	87%	18.4
PET/CT every 3 months (*n* = 43)	97%	92%	92%	19.2
PET/CT every 6 months (*n* = 82)	95%	90%	90%	13.2
CT-Scan, total (*n* = 290)	95%	92%	92%	14.2
PET/CT, total (*n* = 125)	95%	92%	88%	16.5

PFS: progression free survival.

## Data Availability

The data presented in this study are available in this article.

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
