# Peer review of "SEGHI Study: Defining the Best Surveillance Strategy in Hodgkin Lymphoma after First-Line Treatment"

_cancers, 2021, doi:10.3390/cancers13102412_

Round 1
Reviewer 1 Report
The authors performed the SEGHI study (“Seguimiento en Hodgkin por imágenes”), with the intention to retrospectively analyze the various follow-up strategies in multiple GELTAMO centers after first CR in patients diagnosed with HL and concluded that follow- up approaches with imaging in Hodgkin lymphoma provide no benefits for patient survival, and they strongly recommend clinical/analytical surveillance for this group of patients. The results are intriguing. However, there are several issues to be corrected or clarified.
Majors)
In Materials and Methods, Page 2) “The patients were categorized into 5 groups according to the strategy used at the referral 4 center: Clinical group (C/A) (follow-up by medical history, laboratory tests, and physical examination); CT-3 (CT every 3 months); CT-6 (CT every 6 months); PET CT-3 (PET/ CT every 3 months); and PET CT-6 (PET/CT every 6 months).”
How these 5 groups were applied at the referral 4 centers? Was it depended on each doctor, or on each patient? Were the groups prespecified, or established at the time of this retrospective analysis? These should be described.
In Discussion, page 11) “It is striking that the advanced CT-3 group appeared to have a greater number of relapses. This result could have to do with a bias in the study, by which it was decided to perform this surveillance strategy in patients with more aggressive characteristics. “
Another reasonable one is that the strategy of the advanced CT-3 group was actually effective for early detection. I agree with the authors that a bias is more plausible for the reason. There should more biases in this retrospective analysis. Therefore, the conclusions in Abstract and Discussion should be de-emphasized. Also, limitations of this study, including not a prospective randomized phase III study, should be described in Discussion.
Minors)
Page 2 of 12) “Along these lines, a cost-effectiveness study indicated that strategies including radiological images in the surveillance of this pathology are not cost effective and could be detrimental to the 3 patient's quality of life.”
The sentence is unclear.
Reference numbers are double for several references.
A legend for the Table 4 is half missing.
Reviewer 2 Report
In the work, „SEGHI study: Defining the best surveillance strategy in Hodg-2 kin lymphoma after first-line treatment”, the authors Mariana Bastos Oreiro et al. present a nice and informative analysis of a large case series of Hodgkin disease follow-up patients. The work is well done and should be published, however, some aspects might be improved.
The study did only include adult patients, this should be mentioned.
Several lymph node locations can be assessed by ultrasound without radiation exposure, this was not mentioned.
If I understand the work correctly, it was not analyzed exactly which degree of bias was due to the fact that some centers performed a different surveillance strategy for high-risk as compared to low-risk (high-stage versus low-stage) patients. The surveillance strategies, in particular the strategy stratification, should be described more in detail.
For patients treated in a clinical study, the purpose of follow-up is not only to identify relapses, but to confirm remission. So an analysis defining the exact number of false-negative results in a clinical/analytical surveillance strategy would also be helpful.
Another purpose of surveillance is identification of secondary malignancies or late effects of therapy. This aspect should also be mentioned in the discussion.
The analysis of risk of secondary leukemia due to diagnostic irradiation should be weighed against the risk of therapeutic irradiation.
Table 4: please write analytical instead of analytical
Line 231 write risk to instead of riskto
Reviewer 3 Report
This study investigates an important issue. However, some revisions are needed to the text before this could be considered suitable for publication; these are small in number, but the first point below is critical to interpretation.
1. The Abstract states that “no significant difference in survival or progression free survival were found”, but there isn’t a single hypothesis test performed in the manuscript to justify saying this. The authors need to:
1.1. Provide some justification as to why the sample size is appropriate to answer the primary question(s) of interest. You explicitly note that HL has a high cure rate and ultimately only ended up with 68 relapses, with a very small number of total patients in some of the follow-up strategy groups. It’s not clear that the data here is substantial enough for definitive claims like those currently included.
1.2. Back up any statements in regard to no differences between strategies with an appropriate statistical analysis.
1.3. Complement all percentages given by an appropriate measure of uncertainty (e.g., 95% CI).
2. A few typographical/pictorially issues:
2.1. “HL” is used in the Simple Summary when this has not been defined.
2.2. The number “3” appears on line 55 for no reason.
2.3. Figures and Tables are formatted poorly. E.g., “The text continues here” should have been deleted, and there is a second Table 2 that has no content.
2.4. The reference list is formatted incorrectly (the numbering).
2.5. The font is far too small in all Figures. The Tables are also not tables; they’re images.
Round 2
Reviewer 1 Report
I think the manuscript has been revised appropriately.
Author Response
Thank you very much for considering that our review of the paper is appropriate.
Reviewer 3 Report
In this revised manuscript the authors have addressed most of my previous comments. I have two remaining (linked) issues to highlight.
1. It is not clear where the 296 and 207 numbers come from. Are you assuming a two-sided test of 'H0: No difference in mortality'? What test are you assuming would be used? Why are the numbers in the two groups different?
2. The paper in it's current from suffers from an 'absence of evidence is not evidence of absence' issue. If you wish to demonstrate that "clinical follow-up for this group of patients is not associated with higher morality" then you should make the default assumption of a test that it is associated with higher mortality. The CIs given don't answer the "hypothesis of the study" - I note that there still isn't an actual hypothesis test performed (computing and stating a collection of CIs is not the same thing).
Author Response
In this revised manuscript the authors have addressed most of my previous comments. I have two remaining (linked) issues to highlight.
- It is not clear where the 296 and 207 numbers come from. Are you assuming a two-sided test of 'H0: No difference in mortality'? What test are you assuming would be used? Why are the numbers in the two groups different?
- Thank you very much for your comment. We made mistake when classifying the Non-Exposed /Exposed ratio as different from 1. This ratio is equal to 1 and so the n in both groups is the same. Therefore, the n needed in each group would be 200. This has been changed in the text. This was calculated using proportions for cohorts studies.
- The paper in it's current from suffers from an 'absence of evidence is not evidence of absence' issue. If you wish to demonstrate that "clinical follow-up for this group of patients is not associated with higher morality" then you should make the default assumption of a test that it isassociated with higher mortality. The CIs given don't answer the "hypothesis of the study" - I note that there still isn't an actual hypothesis test performed (computing and stating a collection of CIs is not the same thing).
- With an assumed equal efficacy of both arms (clinical and image follow up), a hypothetical inferiority of clinical surveillance with an overall survival rate of 80% versus a rate of 90% or more with image (corresponding to a non-inferiority margin of 10%) had to be excluded with 95% confidence and a power of 80%. This was changed in the text (page 72-82)
Round 3
Reviewer 3 Report
The authors have addressed my previous comments. I have no additional suggestions for revisions.